# Nonspecific Binding of a Putative S-Layer Protein to Plant Cell Wall Polysaccharides—Implication for Growth Competence of *Lactobacillus brevis* in the Gut Microbiota

**DOI:** 10.3390/ijms262311612

**Published:** 2025-11-30

**Authors:** Zhenzhen Hao, Wenjing Zhang, Jianzhong Ge, Daoxin Yang, Kairui Guo, Yuan Wang, Huiying Luo, Huoqing Huang, Xiaoyun Su

**Affiliations:** 1State Key Laboratory of Animal Nutrition, Institute of Animal Sciences, Chinese Academy of Agricultural Sciences, Beijing 100193, China; zhenzhenhao2012@163.com (Z.H.); m15927146115@163.com (W.Z.); gejianzhong163@163.com (J.G.); 821012410058@caas.cn (D.Y.); guokr0712@163.com (K.G.); wangyuan08@caas.cn (Y.W.); luohuiying@caas.cn (H.L.); 2College of Animal Science and Technology, Ningxia University, Yinchuan 750021, China

**Keywords:** Lactobacilli, *Lactobacillus brevis*, PCWPs, S-layer protein, gut bacteria, adhesion

## Abstract

Plant cell wall polysaccharides (PCWPs) serve as an abundant but recalcitrant carbon source for many microbes living in the gut of humans and animals. An adhesion to PCWPs is common in gut bacteria and can even be observed in the lactobacilli, which are supposed to promote the growth competence of these non-PCWP degraders because of the facilitated acquisition of newly released oligosaccharides. Nevertheless, the binding of molecules of lactobacilli to PCWPs and the underlying mechanisms remain largely unknown. By analyzing the transcriptome of *Lactobacillus brevis* grown in xylan supplemented with a xylanase, a gene was identified to encode a putative S-layer PCWP-binding protein (Lb1145). Lb1145 was predicted to have four domains, among which domains 1 and 2 were responsible for binding PCWPs. The binding was nonspecific, since structurally distinct PCWPs, e.g., cellulose, xylan, mannan, and chitin, and even lignin, were all bound by Lb1145. Both of the two N-terminal domains have a high pI, and we demonstrated that a non-enzymatic glycosylation-like process plays an important role in binding. Compared with another *L. brevis* surface protein, i.e., the WxL protein Lb630, Lb1145 displayed a binding preference for the phloem sieve tube in the wheat stem section. Moreover, Lb1145 could bind ten strains within the *Lactobacillus*, *Enterococcus*, *Pediococcus*, and *Bacillus* genera among the seventeen selected gut bacterial species. An analysis of the reported S-layer proteins from the Gram-positive bacteria (lactobacilli and bifidobacteria) and outer membrane proteins from the Gram-negative (*Bacteroides fragilis* and *Prevotella intermedia*) indicated that bacterial cell surface proteins with high pI values are not rare. The high pI-based and non-enzymatic glycosylation-like process-mediated binding represents a new paradigm and may be popular in gut bacterial surface proteins binding to PCWPs, with important physiological implications in growth competition in the gut microbiota.

## 1. Introduction

The dysbiosis of the gut microbiota is well known to be associated with numerous diseases. For example, growing evidence points to the relation of gut microbiota dysbiosis to chronic diseases, the cost of which is estimated to reach USD 47 trillion by 2030 [1,2]. Plant cell wall polysaccharides (PCWPs) serve as an abundant but recalcitrant carbon source for many microbes living in the guts of humans and animals. The catabolism of PCWPs, basically through the fermentation of their derived simple oligosaccharides and monosugars, enables specified gut bacteria to acquire energy in the highly competing intestinal niche [3,4]. Consequently, the thriving of these organisms helps to maintain the homeostasis of the gut, avoid gut microbiota dysbiosis, and ultimately maintain the health of the hosts [5,6,7]. A large portion of the PCWP-utilizing bacteria, such as the *Bacteroides* spp., *Prevotella* spp., and *Ruminococcus* spp., is endowed with the ability to depolymerize recalcitrant polysaccharides [8,9,10,11]. This capability can be ascribed to the orchestrated action of bacterial-encoded glycoside hydrolases (GHs) such as cellulase, xylanase, mannanase, and pectinase [9,11,12,13,14]. Moreover, gut bacteria may also evolve additional strategies, such as the excretion of bacteria-killing agents, the cross-feeding of nutrients, etc., that assist them in gaining more growth competence [15,16]. Remarkably, among the strategies, there is also the adhesion of gut bacteria to PCWPs, which is widely believed to enable the organisms to gain a proximity to bulky polysaccharides for the facilitated assimilation of released oligosaccharides and monosugars. For example, in the PUL (standing for Polysaccharide Utilization Loci) system that is utilized by *Bacteroides* spp. to degrade PCWPs, extracellular cell membrane-attached PCWP-binding proteins (e.g., surface glycan-binding proteins) are often discovered [3,17,18]. The rationale is that, in this way, the oligosaccharides released by the GHs can be rapidly transported into the periplasm for further degradation and subsequent internalization for metabolism, with a lowered chance of loss [13,19,20]. A similar adhesion phenomenon, but with seemingly different underlying mechanisms, has also been noted for other gut bacteria such as *Fibrobacter succinogenes*, *Ruminococcus albus*, and *Roseburia intestinalis* [21,22,23].

Note that, for PCWP-adhesive degrading bacteria, the mode of carbon and energy acquisition is sometimes regarded as “selfish”. This may best be exemplified by the glycoside hydrolases attached to cell membranes in the PUL system of the Bacteroidetes [19]. Despite these adhesive machineries involving enzyme and non-catalytic moieties, it is reasonable to postulate that the “selfish” mechanism does not necessarily mean “exclusive” due to the unavoidable leakage of the released oligosaccharides and monosugars into the surrounding milieu. In addition, many gut bacteria encode glycoside hydrolases without any carbohydrate-binding module (CBM) [24,25,26]. These collectively suggest that there must be considerable amounts of non-immobilized GHs in the gut lumen. Indeed, both free and fiber-attached glycoside hydrolases can readily be detected in the gut [27,28]. Therefore, ample amounts of PCWP-derived dissociative oligosaccharides exist in the gut lumen, supporting the growth of the oligosaccharide utilization bacteria.

Not unexpectedly, lactobacilli and bifidobacteria, the two well-known genera of probiotic bacteria without a PCWP-degrading ability in the gut, are common gut commensals and can grow readily on PCWP-derived oligosaccharides [29,30]. Interestingly, like their PCWP-degrading bacterial counterparts, attaching to PCWPs has also been frequently observed for members in these bacteria, e.g., *L. brevis*, *Lactobacillus casei*, *Lactobacillus plantarum*, and *Bifidobacterium animalis* subsp. *lactis* [31,32,33,34]. This paradox may be explained by the hypothesis that, through adhesion, the chance for absorption of the newly released PCWP-derived oligosaccharides by these gut bacteria should be increased. In addition, adhesion to PCWPs can provide an additional advantage, i.e., stress tolerance. For instance, it has long been observed that lactobacilli adhering to plant fibers can survive in the gastric acid longer, indicating that they have a higher chance to pass the stomach and enter the intestine [32,35].

A common strategy for bacteria to bind to PCWPs is through their cell surface proteins. The best-defined form of PCWP-binding proteins may be the CBMs. For example, the surface proteins encoded by the PUL system of *Bacteroides thetaiotaomicron* have family 6, 35, and 91 CBMs, which allow them to bind strongly to cellulose, mannan, and xylan [36]. Nevertheless, more and more new families of binding proteins, such as the Ig-like domain [37], fibronectin type 3 (Fn3) [38], and Tāpirins [39], etc., are being discovered. Lactobacilli are important probiotic gut commensals, which have tremendous application values in promoting the health of humans and animals. However, the mechanisms underlying the binding of lactobacilli to PCWPs are poorly understood. Recently, we discovered that four predicted surface WxL proteins and one putative S-layer protein are used by *L. brevis* to bind to cellulose and xylan. One unique property of the *L. brevis* WxL proteins is that they have only one domain, in sharp contrast to other characterized homologous proteins. The binding of the identified proteins to PCWPs appears not as strong as canonical CBMs, raising an important question whether there would be additional proteins still contributing to the binding of *L. brevis* to PCWPs [33]. It was noted from transcriptomic analyses that many putative cell surface proteins, in addition to the above-mentioned proteins, were also up-regulated during the growth of *L. brevis* in the MRS medium supplemented with wheat arabinoxylan (WAX) and xylanase (MRS/WAX + Xyn, providing xylooligosaccharides as a carbon source). In this study, Lb1145, one such putative cell surface protein with a high transcription abundance, was identified to bind nonspecifically to PCWPs as well as gut bacteria.

## 2. Results

### 2.1. Identification of Plant Cell Wall-Binding Proteins from L. brevis

PCWPs have been found to bind to a substantial set of substrates, such as lignocellulosic polymers and proteins. By uptaking moisture, the mechanical properties of cellulose and xylan at the nanoscale will be decreased, favoring the gut microbiota attack and digestion in the host organism [40,41]. Herein, the binding of putative surface proteins of *L. brevis* to representative PCWPs was investigated. The *L. brevis* ACCC 63307 strain was isolated from the intestinal content of an Arbor Acres broiler chicken. The genome of this bacterium encodes xylooligosaccharide-degrading genes, enabling it to grow on the MRS medium supplemented with xylooligosaccharides (XOSs) or wheat arabinoxylan (WAX) plus a xylanase. Previously, it was discovered that the strain grown on MRS/WAX + Xyn can bind to crystalline cellulose [33]. Four predicted secreted WxL-domain proteins and one putative S-layer protein were determined to bind to cellulose and/or insoluble WAX [33]. However, whether there are additional surface proteins that can bind to plant cell wall polysaccharides remains unknown.

The transcriptomic data of this *L. brevis* strain grown in MRS/WAX + Xyn and MRS/glucose were compared [33]. Seven more genes, encoding putative cell surface proteins Lb634, Lb636, Lb1145, Lb1181, Lb1345, Lb2328, and Lb2554, induced in MRS/WAX + Xyn (setting 1.5-fold as a threshold) to high transcript abundance were selected for further analysis. These genes were expressed in *Escherichia coli*, and the recombinant proteins were purified through immobilized metal affinity chromatography. Using Avicel and insoluble WAX as representative model PCWPs, it was not unexpected to discover that Lb636, like its homologous WxL proteins (Lb630, Lb631, Lb632, and Lb635) and the LPxTG protein Lb634, could also bind to cellulose and xylan. Lb1145 and Lb2328 appeared to have a strong binding ability with these two substrates. The other proteins, i.e., Lb1181, Lb1345, and Lb2554, could all bind to cellulose and xylan, although the binding abilities were comparably weaker (Figure 1).

### 2.2. Delineation of Lb1145 into Four Domains

Because Lb1145 appeared to bind to cellulose and xylan most strongly and had the highest transcript abundance in the MRS/WAX + Xyn medium, it was selected for further biochemical analyses. An amino acid sequence alignment indicated that Lb1145 has a 63% amino acid sequence identity to the SlpA of *L. brevis* ATCC 8287, which was reported to be a surface layer protein [42]. This suggests that Lb1145 is likely also a cell surface protein. Interestingly, the two proteins are most similar in the N terminal half (1–225 amino acids), having an 88% amino acid sequence identity. When the recombinant Lb1145-DsRed fusion protein was incubated with *L. brevis* cells, fluorescence microscopic observations clearly showed red fluorescence on the bacterial cells, while no signal was observed with the DsRed control alone (Appendix A). Given that the cell surface proteins of lactobacilli often self-assemble, this suggested that the observed fluorescence might be a result of the recombinant Lb1145-DsRed interacting with the native Lb1145 on the cell surface, serving as the anchoring point. However, there is also the possibility that exogenously added Lb1145-DsRed bound to unidentified components on the cell surface. Therefore, the location of Lb1145 on the cell surface needs further investigation.

Unlike the WxL proteins in *L. brevis*, which have a single domain, the AlphaFold 3 analysis of Lb1145 predicted with high fidelity (pTM score = 0.62) that this protein might contain four domains (Figure 2A). From the AlphaFold 3 analysis, the boundaries of the four domains could be delineated as follows: domain 1, 31–136 aa (the first 30 aa are the predicted signal peptide); domain 2, 137–225 aa; domain 3, 226–325 aa; and domain 4, 326–462 aa. Interestingly, the superimposition of the first two domains (with an amino acid sequence similarity of 34.78%) indicated that they were structurally similar to each other (RMSD = 2.43 Å) (Figure 2B).

For a detailed analysis of the binding properties, the genes encoding the four domains were individually cloned into pET-28a(+). Although the full-length protein without the signal peptide (denoted WT), domain 3, and domain 4 were successfully expressed in *E. coli*, no perceivable expression was observed for domains 1 and 2. However, when the gene coding for the red fluorescence protein DsRed was fused C-terminal to domains 1 and 2, the truncation mutants could be successfully obtained. In addition, the mutant including the tandem linked domains 1 and 2 could also be recombinantly produced. These results suggested that the two domains were probably prone to proteolytic degradation during the folding process and an expression partner thus assisted in their successful folding. Thus, domains 1-DsRed and 2-DsRed were used for subsequent biochemical analyses. The truncated proteins of domain 1 (in fusion with DsRed), 2 (in fusion with DsRed), 3, 4, and 1 + 2 were denoted TM1 (standing for truncation mutant 1), TM2, TM3, TM4, and TM5, respectively (Figure 2C), and were recombinantly produced and purified, as analyzed using SDS-PAGE (Figure 2D).

### 2.3. The Two N-Terminal Domains of Lb1145 Are Responsible for Nonspecific Binding

The wild-type Lb1145 and its truncation mutants were determined for their abilities to bind to three major plant PCWPs (cellulose, xylan, and mannan). Chitin and lignin, representing a major common polysaccharide of fungi and the non-polysaccharide polymeric component of a plant cell wall, respectively, were also used as two substrates. Interestingly, in addition to crystalline cellulose Avicel and insoluble WAX, the Lb1145 WT could bind to mannan, chitin, and lignin as well (Figure 3A). Both TM1 and TM2 also bound to the PCWPs, but to a much lower extent. The DsRed did not bind to these polysaccharides, indicating that the binding was from the interaction of domains 1 and 2 to the substrates. In contrast, neither domain 3 nor domain 4 bound to the polysaccharides, further supporting the notion that the binding of Lb1145 to the PCWPs was essentially from domains 1 and 2. TM5, bearing both domains 1 and 2, appeared to bind to the PCWPs more strongly (Figure 3A). Quantitating the binding constants of WT, TM1, and TM2 to the Avicel cellulose by depletion binding isotherms was not successful because part of the proteins appeared to stick to the tubes, preventing an accurate determination of cellulose-bound and unbound proteins.

### 2.4. Binding of Lb1145 to the Wheat Stem

The WxL proteins from *L. brevis* could also bind to PCWPs such as cellulose and xylan. The amino acid residues, as well as the predicted structures, of WxL proteins and Lb1145 are very different. It was interesting to learn whether there would be a difference regarding the binding characteristics of the two kinds of proteins. Wheat stem, which is rich in lignocellulosic compounds, is used as a representative natural and complex PCWP substrate to determine the binding of Lb1145 and a previously characterized WxL protein Lb630. In comparison with wheat stem, Avicel cellulose, wheat arabinoxylan, mannan, and pectin are all purified and isolated model plant cell wall polysaccharides. Therefore, wheat stem is more appropriate than these PCWPs to serve as a natural and complex model substrate in the intestine. In addition, checking the fluorescence of the fused DsRed/GFP proteins would allow us to intuitively perceive if there is a difference in the binding preference of Lb1145 and other putative cell surface proteins. Therefore, the fusion expression of Lb1145 with DsRed, and Lb630 with EGFP, was carried out, and the two chimeric proteins were recombinantly expressed and purified. While the DsRed and EGFP controls did not have a significant binding to wheat stem slices, both Lb1145-DsRed and Lb630-EGFP could bind well to the wheat slices. However, there were perceivable differences in the binding sites and binding strength: Lb1145-DsRed bound mainly to the mechanical tissue and vascular bundle, while Lb630-DsRed could also bind the abundant parenchyma fibers in addition to these (Figure 3B, upper panel). In the enlarged view, it could also be seen that Lb1145-DsRed bound more strongly to the phloem sieve tube, while Lb630-EGFP preferred to bind to the parenchyma among the xylem vessels (Figure 3B, lower panel).

### 2.5. Involvement of a Non-Enzymatic Glycosylation-like Process in Binding of Lb1145 to PCWPs

A Dali-server analysis indicated that the predicted structures of Lb1145 domain 1 and domain 2 did not resemble any known Type 1, 2, and 3 CBMs or other PCWP-binding proteins [43]. For CBMs, binding to crystalline cellulose normally involves a flat surface of the protein with regularly arranged aromatic or polar acid residues that interact with cellulose via hydrophobic packing or hydrophobic bonding. However, neither a flat surface nor such regularly arranged aromatic/polar residues were discovered in the first and second domains. Moreover, given that TM1 and TM2 have such a wide substrate promiscuity and that cellulose, xylan, mannan, chitin, and lignin are structurally very divergent, it appears unlikely that the two first domains bind to all these substrates essentially through hydrophobic interactions and/or hydrogen bonding. This is because the substrate specificity, as revealed for CBMs, is highly specific and determined by the structural compatibility of the protein and substrates. Therefore, a mechanism involving nonspecific binding is instead more likely involved in the binding of the two domains to these substrates.

It was noted that, while the calculated pI of the full-length Lb1145 was 9.51, those of the two non-binding truncation mutants (TM3 and TM4) were 4.39 and 4.83, respectively. In contrast, domain 1 and domain 2 had much higher pI values (10.07 and 9.67, respectively), thus making the most important contribution to the alkalic pI of the overall protein of Lb1145. Heterogeneous PCWPs, e.g., xylan, can be negatively charged through side chain modifications (such as glucuronic acid), serving as an ideal binding substrate for high pI proteins such as WT, TM1, TM2, and TM5 via electrostatic interaction. However, glucose, the monomer of homogeneous cellulose, can also spontaneously form complexes with proteins (e.g., human serum albumin), which is known as non-enzymatic glycosylation [44]. This process involves the hydrogen bonding and electrostatic attraction of glucose with the lysines in the proteins, incurred through the stretching of the N-H bond belonging to the NH3^+^ group of lysine along an oxygen atom of glucose [45]. Domains 1 (106 aa in length) and 2 (89 aa in length) have 14 Lys/5 Arg and 9 Lys/2 Arg residues, respectively, much more abundant than those (3 Lys/1 Arg and 8 Lys/1 Arg) in domains 3 (100 aa in length) and 4 (137 aa in length). The non-decorated cellulose, xylan, and mannan are composed of hundreds of such monosugars. Therefore, even for these non-decorated PCWPs, the abundant hydroxyl and hemiacetal groups may readily allow interactions with the plentiful positively charged residues (Lys and Arg) in domains 1 and 2 via a non-enzymatic glycation-like process. Indeed, 0.12 ± 0.03 μM of protein–glucose Amadori products was detected for 10 μM of TM5 incubated with 1 M of glucose (the monomer of cellulose) for 1 h, whereas no such product could be observed for TM3 and TM4.

The non-enzymatic glycosylation is a slow and gradually developing process in which hydrogen bonding and electrostatic interaction can be involved [45]. Despite the observation of protein–glucose Amadori products, no such products were detected in the binding of Lb1145 to cellulose and xylan, suggesting that the hydrogen bonding and electrostatic interaction-based intermediate, but not the chemically linked end-point reaction products, predominated in the binding. These interactions are well known to be able to be disrupted by high concentrations of ions. Therefore, the effect of increasing ion concentrations in the buffer on the interaction of Lb1145 with cellulose was evaluated. Although the relative binding abilities of TM5 ranging from (81.4 ± 0.79)% to (100.0 ± 1.72)% were highly comparable when the buffer contained 0, 75, 150, and 300 mM of NaCl, the binding abilities largely decreased to (44.0 ± 1.0)% and (38.0 ± 1.9)%, respectively, when the concentration of the salt was increased to 600 mM and 1.2 mM. The binding even disappeared completely when the salt concentration increased to 2.4 M (Figure 4A). This strongly indicated that hydrogen bonding and/or electrostatic attraction was involved in the binding of Lb1145 to the polysaccharides.

Varying the buffer from 5.0 to 7.0 had a trend to decrease the relative binding from (44.7 ± 0.7)% to (31.1 ± 0.1)% (*p* = 0.071). The increase in pH was supposed to decrease the positive charges in TM5, which might be responsible for the apparent reduced binding. However, changing the buffer from 7.0 to 10.0 largely increased the binding from (31.1 ± 0.1)% to (100.0 ± 1.1)% (Figure 4B). In alkalic solutions, the enhanced disassembly of Avicel into more nano-fibers with an elevated interacting surface has been observed [46]. Even in solutions with a pH identical to that of a protein, there are also amino acid residues that can harbor positive charges. These include arginine, with a pKa of 13.8 [47], and lysine, with a pKa of ca. 10.53 [48]. Therefore, the apparent improved binding ability at pH 9.0 and 10.0 could be an overall effect from the elevated cellulose nano-particle surface area overwhelming the negative effect incurred by the decrease in positive charges on the protein.

### 2.6. Binding of Lb1145 to Gut Bacteria

The animal intestine is a crowding niche that contains billions of bacteria, suggesting that there is a chance that gut bacteria can physically contact each other. Indeed, one means for the frequently observed horizontal gene transfer in the gut bacteria is through conjugation, which relies on close contacts between cells of the same or of different species [49]. The ability of Lb1145 to bind to selected bacteria was evaluated. Lb1145 could bind to *L. brevis* and some other *lactobacilli*, such as *Lactobacillus crispatus*, *Lactobacillus johnsonii*, and *Lactobacillus salivarius*, but not *Lactobacillus mucosae* and *Lactobacillus reuteri* (Figure 5A). In addition, it could bind to *Enterococcus durans*, *Enterococcus faecalis*, *Pediococus pentosaceus*, *Bacteroides ovatus*, and two *Bacillus* strains (*Bacillus subtilis* and *Bacillus licheniformis*). However, no binding was observed for *Pediococus acidilactici*, *Bifidobacterium pseudolongum*, *Bifidobacterium longum*, *E. coli*, and *Bacteroides uniformis* (Figure 5A). Note that the binding was highly variable, with the weakest binding rate of (11.9 ± 0.8)% monitored for *E. faecalis* and the highest binding rate of (43.3 ± 0.9)% for *B. subtilis* (Figure 5A).

### 2.7. Lb1145 Partially Restored the Adhesion of LiCl-Treated L. brevis to the Intestinal Epithelial Cell IPEC-J2

LiCl treatment can remove the surface layer proteins of Gram-positive bacteria [50], thereby decreasing the adhesion of a bacterium to intestinal epithelial cells. Indeed, treating *L. brevis* with 5M LiCl significantly lowered the adhesion rate of this bacterium from (67.0 ± 2.4)% to (58.5 ± 0.7)% (*p* < 0.01). Treating *L. brevis* with 1 μM of Lb1145 partially restored the adhesion rate to (64.4 ± 1.1)% (Figure 5B). These findings strongly suggested that the surface layer proteins such as Lb1145 must play an important role in enhancing the adhesion of the *L. brevis* to gut epithelial cells.

## 3. Discussion

Previously, the S-layer proteins of lactobacilli have been demonstrated to play an important role in the adhering of the bacteria to host cells [50,51]. For example, the C-terminal part of SlpA is highly conserved in species of *Lactobacillus acidophilus* and displays lectin-like activity [52,53]. In contrast, although binding to PCWPs is a common trait observed for many gut-resident bacteria, most of the underlying mechanisms remain unknown. Herein, the in vitro biochemical analyses indicated that seven putative surface layer proteins of *L. brevis*, additional to the five reported earlier [33], were able to bind to PCWPs, among which Lb1145 and those belonging to other families are included. Therefore, this finding unequivocally suggested that, in addition to the well-known canonical CBMs, there are other yet-to-characterize kinds of proteins able to bind PCWPs.

CBMs, despite being promiscuous for some of them, are usually specific to only limited kinds of polysaccharides with a close structural similarity. For example, two family 16 CBMs (CBM16-1 and CBM16-2 from *Caldanaerobius polysaccharolyticus*) have a binding promiscuity to both cellulose and mannan. Glucose (the monomeric component of cellulose) and mannose (the monomer of mannan) are only different at the C-2 hydroxyl (equatorial for glucose but axial for mannose). In this case, both of them can be accommodated by the finely tuned key aromatic and polar residues from the two CBMs [54]. Therefore, the restricted substrate scope is an inherent nature of CBMs because their interaction with PCWPs stems essentially from amino acid-based hydrophobic stacking and/or hydrogen bonding. In contrast, Lb1145 can bind to different PCWPs and even non-sugar-based aromatic polymer lignin with largely different chemical compositions and structures. By dissecting Lb1145 into four domains based on structural prediction, we were able to attribute the binding ability to its two N-terminal domains. Further analyses indicated that the high pIs of these two domains could be responsible for the binding.

The other five putative *L. brevis* PCWP-binding surface proteins in our previous reports (Lb630, Lb631, Lb632, Lb635, and Lb1325) had predicted pIs of 4.58, 4.64, 4.70, 4.41, and 9.51, respectively [33]. The other seven proteins in this study (Lb634, Lb636, Lb1181, Lb1345, Lb2328, and Lb2554) had predicted pIs of 9.20, 9.55, 9.95, 5.28, 10.42, 9.85, and 4.91, respectively. This suggested that surface PCWP-binding proteins with high pI values might be a common phenomenon for certain gut bacteria. As Lb1145 is a putative S-layer surface protein from the Gram-positive bacterium *L. brevis*, it is thus interesting to know if other S-layer proteins, as well as other cell surface proteins even from Gram-negative bacteria, would have similar high pI domains, enabling them to interact nonspecifically with PCWPs. Therefore, the four most commonly observed gut commensal genera, including two Gram-positive (the lactobacilli and bifidobacteria) and two negative (the *Bacteroides* and *Prevotella*) ones, were selected as representative gut bacteria for bioinformatics analysis. The reported 207 S-layer proteins from lactobacilli and bifidobacteria and 35 outer membrane proteins from *Bacteroides fragilis* and *Prevotella intermedia* 17 were collected for the prediction of their pIs and the delineated domains. It appeared that the surface proteins with high pIs (herein, “>9.0” was arbitrarily set as the threshold) were indeed not rare. For example, the bifidobacteria have 5 (among 48 analyzed) such proteins with pIs ranging from 9.18 to 10.02, the lactobacilli have 113 proteins (among 159 analyzed) with pIs ranging from 9.05 to 10.56, the *Bacteroides* have 3 proteins (among 12 analyzed) with pIs ranging from 9.02 to 9.45, and the *Prevotella* have 4 proteins (among 23 analyzed) with pIs ranging from 9.12 to 9.38, respectively (Appendix A). Note that all these four bacteria have been documented to be able to utilize PCWP-derived oligosaccharides [29,55,56,57,58,59]. Therefore, whether the identified high pI cell surface proteins could similarly bind to PCWPs (as well as other gut bacteria and epithelial cells) is an intriguing question.

Interestingly, similar high pI extracellular proteins PWBP57 and PWBP65 with a PCWP-binding ability have been discovered in a thermophilic bacterium *Caldicellulosiruptor bescii* [60]. The two proteins are 500 (for PWBP57) and 556 (for PWBP65) amino acids in length and have a highly conserved structure. Neither of the two N-terminal domains of Lb1145 shares a high amino acid sequence homology or structural similarity to any of these two thermophilic bacterial proteins. Given the fact that the two kinds of protein are discovered in completely different niches (animal gut versus hot spring), it can be deduced that these two different kinds of proteins from divergent origins evolved to functional convergence.

Although the binding constants of Lb1145 and its N-terminal domains could not be accurately quantitated, it could be perceived from the gel (Figure 1 and Figure 3A) that the binding ability of Lb1145 to insoluble PCWPs is lower than that of typical CBMs [61,62]. This might help to explain why *L. brevis* utilizes many, rather than a few, putative cell surface proteins in binding. Moreover, although the fluorescence signals for Lb1145 and the WxL protein Lb630 bound to the wheat stem section were majorly overlapping, there were slight but perceivable differences in their preferred binding area. This suggests that the discovered putative surface binding proteins have complementary functions. Therefore, *L. brevis* may benefit from the expression of multiple putative PCWP-binding cell surface proteins, since their combination may help the bacterium compete in the intestinal environment. Similar but divergent binding functions have also been clearly demonstrated for some CBMs, even within the same family (such as the family 35 members) [63].

The functions of Lb1145 are apparently more than just binding to PCWPs, as it can also bind to some other gut bacteria in addition to *L. brevis*. Bacteria-to-bacteria communication can happen through excreted small molecules such as quorum sensing signals [64,65], but direct cell contacting is also a common means, as exemplified by the widely encountered cell-to-cell conjugation. It is, therefore, hypothesized that the binding of the putative surface layer protein Lb1145 to other gut bacteria may also promote such communication between the same or even different species of bacteria.

With the current discoveries, however, it is not known if additional functions can be assigned to the binding proteins. However, for Lb1145, the chance is large because Lb1145 has two additional C-terminal domains whose functions remain to be elucidated. For SlpA, the C-terminal domains are supposed to be involved in self-assembly [66]. Among the identified proteins, similar to Lb1145, Lb1181, Lb1345, Lb2328, and Lb2554 also have multiple domains. Whether the other unidentified functions will similarly have an implication to benefit *L. brevis* to compete in the gut niche remains to be elucidated. For the high pI proteins from lactobacilli, bifidobacteria, *B. fragilis*, and *Prevotella intermedia* (Appendix A), the majority of them (86 among the 128 proteins) were also predicted to have multidomains (from two to six). In most of the multimodular proteins, the high pI of the full-length protein was attributed to only part of the domains (except for AAV42087.1, AAV42860.1, AAV42037.1, AAV43418.1, Q5FM84.1, A0A856KME7, C2KBR4, C2KDQ5, A8YX60, and J4BP74). Although the exact roles of these proteins are often not known, some appear to have definite functions, e.g., the TonB-dependent transporters on the outer membrane of *B. fragilis*. Therefore, if the high pI proteins were determined to also bind to PCWPs, these proteins might be regarded as moonlighting proteins with multiple physiologically relevant properties within one polypeptide chain [67].

Based on the current findings and our previous research [33], a model can be proposed to describe how *L. brevis* might adhere to PCWPs, other gut bacteria, and intestinal epithelial cells with the assistance of the putative cell surface protein Lb1145 and the others. The adherence, basically through a nonspecific mode of action, is hypothesized to assist *L. brevis* to thrive in the highly competitive gut niche by acquiring newly released oligosaccharides in close proximity to the recalcitrant PCWPs (in addition to the dissociative fermentative sugars) and exchanging information with other gut bacteria and the host epithelial cells (Figure 6).

## 4. Materials and Methods

### 4.1. Microbial Strains and Culture

The *E. coli* Trans1-T1 and *E. coli* BL21(DE3) strains (Transgen, Beijing, China) were used for plasmid construction and recombinant protein expression, respectively. All *E. coli* strains were grown in Luria–Bertani (LB) medium at 37 °C. The facultative anaerobic bacterium *L. brevis* ACCC 63307, *L. mucosae* ACCC 63411, *L. crispatus*, *L. johnsonii*, *L. reuteri*, *L. salivarius*, *E. durans* ACCC 63310, *E. faecalis* ACCC 63309, *P. pentosaceus* ACCC 63311, and *P. acidilactici* ACCC 63312 were all isolated from the intestine of broiler chickens and cultivated in the de Man, Rogosa, and Sharpe (MRS) medium at 37 °C. *B. pseudolongum*, *B. longum*, *B. uniformis*, and *B. ovatus* were isolated from the intestine of broiler chickens and cultured in the Glfu anaerobic medium (GAM) (proteose peptone, 10.0 g/L; soybean peptone, 3.0 g/L; proteose peptone, 10.0 g/L; digestive serum powder, 13.5 g/L; beef powder, 2.0 g/L; yeast extract, 5.0 g/L; glucose, 3.0 g/L; bovine liver extract powder, 1.2 g/L; sodium chloride, 3.0 g/L; soluble starch, 5.0 g/L; potassium dihydrogen phosphate, 2.5 g/L; L-cysteine, 0.3 g/L; sodium thioglycolate, 0.3 g/L; final pH 7.1 ± 0.2). All these bacteria were cultured anaerobically for 2 d. The *E. coli L13*, *B. subtilis L2*, and *B. licheniformis* L1 were isolated from the feces of broiler chickens and were cultured in the LB medium aerobically with shaking at 37 °C overnight.

### 4.2. Gene Cloning, Expression, and Protein Purification

The nucleotide sequences of the primers used to amplify the genes in this study are detailed in Appendix A. The PCR amplification steps used were as follows: an initial pre-denaturation at 98 °C for 10 min, followed by 34 cycles of 98 °C for 30 s, 65 °C for 30 s, and 72 °C for a certain period depending on the length of the amplified DNA fragments (1 min/kb), with the final extension at 72 °C for 10 min. The genes encoding Lb634 (GenBank accession number XQO30360.1), Lb636 (GenBank accession number XQO30361.1), Lb1145 (GenBank accession number XQO30362.1), Lb1181 (GenBank accession number XQO30363.1), Lb1345 (GenBank accession number XQO30364.1), Lb2328 (GenBank accession number XQO30365.1), Lb2554 (GenBank accession number XQU59201.1), Lb1145TM1, Lb1145TM2, Lb1145TM3, Lb1145TM4, and Lb1145TM5 were amplified from the genomic DNA of *L. brevis* and cloned into the *Nde*I/*Not*I-digested pET-28a(+) plasmid to obtain the corresponding plasmids (pET28a-Lb634, pET28a-Lb636, pET28a-Lb1145, pET28a-Lb1181, pET28a-Lb1345, pET28a-Lb2328, pET28a-Lb2554, pET28a-Lb1145TM1, pET28a-Lb1145TM2, pET28a-Lb1145TM3, pET28a-Lb1145TM4, and pET28a-Lb1145TM5, respectively). The *DsRed* and *egfp* genes encoding the coral red fluorescence protein and enhanced green fluorescence protein were amplified from the pCbh1-DsRed [68] and the pEGFP-N1 [69] plasmids, respectively. The genes coding for the full-length Lb1145 and its truncation mutants TM1 and TM2 were individually assembled with the *DsRed* gene and the *Nde*I/*Not*I linearized pET-28a(+) plasmid using the Gibson assembly method to obtain pET28a-Lb1145-DsRed, pET28a-TM1-DsRed, and pET28a-TM2-DsRed [70]. The gene encoding Lb630 was assembled with the *egfp* gene and the pET-28a(+) plasmid to obtain pET28a-Lb630-egfp. These plasmids were individually transformed into *E. coli* BL21(DE3) for recombinant expression and purification using a method essentially the same as described previously [33]. Briefly, the BL21(DE3) transformants containing each of the plasmids were cultured in 50 mL of LB supplemented with 50 µg/mL of kanamycin overnight at 37 °C. The pre-culture was transferred into 350 mL fresh LB medium (2% *v*/*v*), and the culture was continued until the OD_600_ reached 0.6–0.8. Isopropyl-β-D-thiogalactopyranoside (IPTG) was added to the culture at a final concentration of 0.5 mM to induce the expression of recombinant proteins. The culture was then incubated at 16 °C for 15 h. The cells were collected through centrifugation and re-suspended in a binding buffer (20 mM Tris-HCl, pH 7.4, 500 mM NaCl). The bacterial suspension was sonicated to break down the cell walls, with centrifugation for 10 min. The supernatant was harvested and passed through a Ni-affinity column resin. The bound proteins were eluted using the elution buffer with varying concentrations of imidazole (50 to 300 mM). The protein elution was analyzed using 15% (wt/vol) SDS-PAGE electrophoresis. The fractions containing the purified proteins were combined, concentrated, and changed to a protein storage buffer (50 mM Tris-HCl, pH 7.4, 150 mM NaCl).

### 4.3. Monitoring Binding of 1145 to L. brevis Using a Confocal Microscope

The *L. brevis* was inoculated into MRS liquid medium and cultured anaerobically for 36 h. The bacterial cells were harvested by centrifugation, washed three times with phosphate-buffered saline (PBS), and then incubated with 1 mg/mL BSA for 0.5 h at room temperature to inhibit nonspecific binding. Following another centrifugation, the cells were washed three times with PBS and re-suspended in equal volumes of Lb1145-DsRed and DsRed, respectively, which were adjusted to the same fluorescence intensity. After incubation at room temperature for 2 h, the binding complexes were washed three times with PBS and observed using a Leica TCS SP8 laser confocal microscope (Leica Microsystem, Wetzlar, Germany).

### 4.4. Binding of the Recombinant L. brevis Putative Surface Proteins to Insoluble PCWPs, Lignin, and Chitin

The binding of the recombinant *L. brevis* putative surface proteins to insoluble PCWPs and lignin was measured as described earlier [33]. Briefly, 100 μg each of the recombinant proteins was mixed with 30 mg of different substrates in the protein storage buffer, with a total volume of 0.4 mL. Specifically, recombinant Lb634, Lb636, Lb1181, Lb1345, Lb2328, and Lb2554 were mixed with Avicel (crystalline cellulose) and insoluble wheat arabinoxylan (IWAX), respectively. Lb1145, its truncation mutants (TM1, TM2, TM3, TM4, and TM5), and DsRed (as a control of TM1 and TM2) were mixed with Avicel (crystalline cellulose), insoluble wheat arabinoxylan (IWAX), mannan, alkali lignin, and chitin, respectively. The mixtures were incubated at 4 °C for 1 h, with periodical mixing by gentle pipetting. The binding mixture was centrifuged to separate the polysaccharides and lignin and the proteins. The precipitates were washed with 1 mL of protein storage buffer four times and re-suspended in 70 μL of SDS-PAGE loading buffer, and then boiled for 5 min to release the bound protein. One tenth the volume of the supernatant (unbound protein) and two tenths the volume of the precipitate (bound protein) were further analyzed through 15% (wt/vol) SDS-PAGE electrophoresis, with the exception of TM3, TM4, and DsRed, for which only one thirtieth of the supernatant volume was used.

### 4.5. Measuring the Protein–Glucose Amadori Conjugate Using a Nitroblue Tetrazolium Method

The protein might form an Amadori conjugate with the monomers of the polysaccharides through a non-enzymatic glycosylation-like process. Such a protein–glucose Amadori product can reduce nitroblue tetrazolium chloride to form formazan, which can be detected at 550 nm. Therefore, glucose, the monomer of cellulose, was used as a model substrate, and its binding to the Lb1145 TM5, as well as TM3 and TM4, was determined using the method described, with slight modifications [71]. Briefly, 10 μM each of the truncation mutants TM3, TM4, and TM5 was individually incubated with 1 M of glucose in 50 mM Tris-HCl, at pH 7.4 and at room temperature for 1 h. The residual glucose was removed by passing the sample through a desalting column. Then, 50 μL of the bound protein was incubated with 150 μL of 2 mM nitroblue tetrazolium chloride solution (dissolved in 0.1 M Na_2_CO_3_/NaHCO_3_ solution, pH 10.35) (Solarbio, Beijing, China) at 37 °C for 20 min. The OD_550_ was measured and the protein–Amadori concentration was calculated based on the 1-deoxy-1-morpholino-D-fructose standard (Maclin, Shanghai, China).

### 4.6. Effect of NaCl and pH on the Binding of the Recombinant Lb1145 TM5 to Cellulose

To determine whether the pH might have an effect on the binding of the Lb1145 TM5 to cellulose, 100 μg of recombinant TM5 and 30 mg of Avicel were incubated in the following buffers in a total volume of 0.5 mL: 50 mM McIlvaine buffer (pH 5.0–6.0), Tris–HCl (pH 7.0–9.0), and glycine–NaOH (pH 9.0–10.0). To assess the effect of NaCl concentration on the binding, the recombinant protein and Avicel were incubated in 50 mM Tris-HCl buffer (pH 7.4) with different concentrations of NaCl (0, 75, 150, 300, 600, 1200, 2400 mM) and then used for the analysis of the relative binding rates.

### 4.7. Binding of the Recombinant Putative L. brevis Surface Proteins to the Wheat Stem Tissue

The stem tissue of growing wheat was cut off and promptly fixed with ethanol–formalin–acetic acid (Coolaber, Beijing, China). Sections of stem tissue (4 μm thick) were prepared and subjected to preprocessing and fluorescence staining as described in the reference [60]. Briefly, the sections were dewaxed twice with xylene firstly. Then, they were rehydrated step by step in different concentrations of alcohol (anhydrous ethanol, 95% ethanol, 90% ethanol, 80% ethanol, 70% ethanol, 50% ethanol, and PBS buffer). After rehydration, the slides were transferred into a pre-boiled citrate antigen retrieval solution and boiled for 10 min. They were then allowed to cool to room temperature slowly and were washed once with PBS buffer (pH 7.4). Then, the slices were subjected to a closure process by incubating them with 3% BSA for 0.5 h at room temperature to inhibit nonspecific binding. The slices were washed with PBS three times and then incubated with an identical fluorescence intensity of Lb1145-DsRed or Lb630-EGFP fusion proteins in the protein storage buffer for 0.5 h. The recombinant DsRed and EGFP were used as the control. Next, the slices were washed with the protein storage buffer four times. The excess buffer was removed with filter paper. The slices were carefully sealed with the fluorescence decay resistant medium. The localization of the proteins on the wheat stem slice was visualized under the Leica TCS SP8 laser confocal microscope (Leica Microsystem, Wetzlar, Germany).

### 4.8. Binding of Lb1145 to Selected Intestinal Bacteria

Seventeen representative gut bacteria isolated from the gut of chickens were used to determine whether they could be bound by Lb1145. The bacteria were cultured in their respective media, collected through centrifugation, and re-suspended with PBS (pH7.4) to the OD_600_ of 1.5. The cell suspension (1.0 mL) was centrifuged, and the supernatant was removed. The cell precipitation and 250 uL recombinant Lb1145-DsRed protein or DsRed (as a control) were mixed, respectively. The mixtures were shaken end-over-end for 0.5 h at 37 °C. After centrifugation, the fluorescence intensity of the supernatant was measured, allowing for calculations of the binding ability of the proteins to intestinal bacteria. The adhesion rate of the bacteria to Lb1145-DsRed was calculated as follows: 100% × (a − b − c)/(a − c), where “a” represents the value of the control (Lb1145-DsRed but no bacteria), “b” represents the fluorescence intensity value of the supernatant of bacteria incubated with Lb1145-DsRed, and “c” represents the fluorescence intensity value of the supernatant of bacteria incubated with DsRed.

### 4.9. Determining Adhesion of L. brevis to the Intestinal Epithelial Cells IPEC-J2

The IPEC-J2 cell line derived from the jejunum of a suckling pig was used to determine the binding ability of *L. brevis* to the intestinal epithelial cells. The IPEC-J2 cells were cultured at 37 °C in Dulbecco’s Modified Eagle’s Medium (DMEM) (Gibco, Waltham, MA, USA) with high glucose, containing 10% fetal bovine serum (FBS) (Sigma-Aldrich, St. Louis, MO, USA). *L. brevis* was cultured in 100 mL of MRS liquid medium anaerobically at 37 °C for 2 d. Then, the culture was divided into two equal parts. The bacterial cells were collected through centrifugation at 6450× *g* for 10 min and then washed with PBS three times. The cell pellets were re-suspended in 2.5 mL of 5 M LiCl solution or PBS (as a control), respectively, and the mixtures were shaken end-over-end for 0.5 h at 37 °C. Then, the bacteria were precipitated through centrifugation. The cell precipitates were washed with PBS three times and re-suspended with 1 μM of Lb1145 protein in a total volume of 1 mL. The mixtures were shaken end-over-end slowly for 1 h at 37 °C. The bacteria were centrifugated and washed with PBS. The bacteria were divided into three equal parts again. To determine the adhesion rate of *L. brevis* to IPEC-J2, 1 mL each of the bacterial suspensions was added to the IPEC-J2 cells (1.0 × 10^6^). The mixture was incubated at 37 °C for 2 h and the unbound bacteria were pipetted out. The cells were washed with PBS three times and digested with 0.5 mL of 0.25% pancreatic enzyme (Maclin, Shanghai, China) for 5 min. Then, the digestion was terminated by the addition of 1 mL of complete culture medium. The mixture containing the epithelial cells and bacteria was centrifuged and re-suspended with 1 mL of PBS. The bacteria adhering to the cells were quantitated by spreading the serially diluted IPEC-J2/*L. brevis* cell suspension on the MRS plates and then counting the colonies. The adhesion rate was calculated by using the following formula: N_1_/N_0_ × 100% (N_1_: with IPEC-J2 cell; N_0_: no IPEC-J2 cell).

### 4.10. Statistical Analysis

The results of each test were shown as a mean ± standard deviation (mean ± SD). One-way analysis of variance (ANOVA) in SPSS 19.0 (SPSS Inc., Chicago, IL, USA) was used for data analysis. Tukey’s post hoc test for multiple comparisons was used to compare the three treatment groups. Significance is denoted by asterisks, with * indicating *p* < 0.05 and ** indicating *p* < 0.01; “ns” signifies that the results are not statistically significant.

### 4.11. Data Availability

The raw transcriptomic data involved in this work is deposited in the National Microbiology Data Center (NMDC) with accession number NMDC10019849 (https://nmdc.cn/resource/genomics/project/detail/NMDC10019849 (accessed on 25 November 2025)).

## 5. Conclusions

We discovered that many of the putative cell surface proteins of *L. brevis* with a high transcription abundance can bind to cellulose or xylan. The representative protein, Lb1145, was delineated into four domains, with the first two having high pIs determined to be responsible for PCWP binding. A non-enzymatic, glycosylation-like process was proposed to likely play a crucial role in this binding. Compared to Lb630, Lb1145 exhibits a different binding preference for the phloem sieve tubes in wheat stem tissues. Furthermore, Lb1145 can bind to bacteria in Lactobacillus, Enterococcus, Pediococcus, and Bacillus. We also discovered that cell surface proteins with high isoelectric points are not uncommon in previously reported Gram-negative and Gram-positive bacteria, pointing to a widespread PCWP-binding paradigm in the gut microbes.

## Figures and Tables

**Figure 1 ijms-26-11612-f001:**
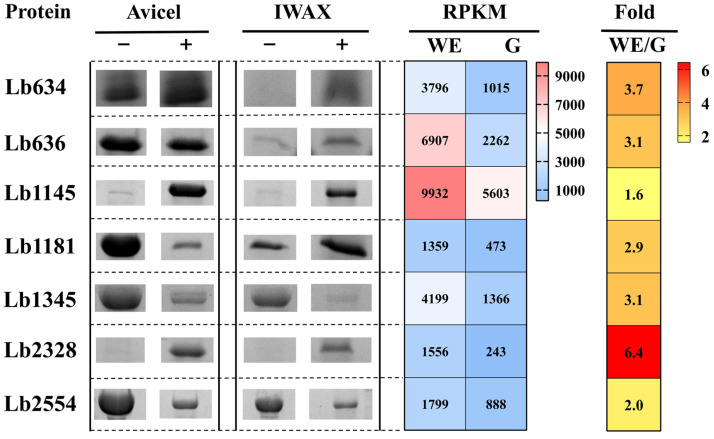
Binding of putative cell surface proteins with high transcript abundance to crystalline cellulose and insoluble WAX. WE: *L. brevis* cultured in the MRS medium supplemented with wheat arabinoxylan and the enzyme xylanase; G: *L. brevis* cultured in the medium supplemented with glucose; Avicel: crystalline cellulose; IWAX: insoluble wheat arabinoxylan; RPKM: Reads Per Kilobase of exon model per Million mapped reads. “−”: unbound protein; “+”: bound protein.

**Figure 2 ijms-26-11612-f002:**
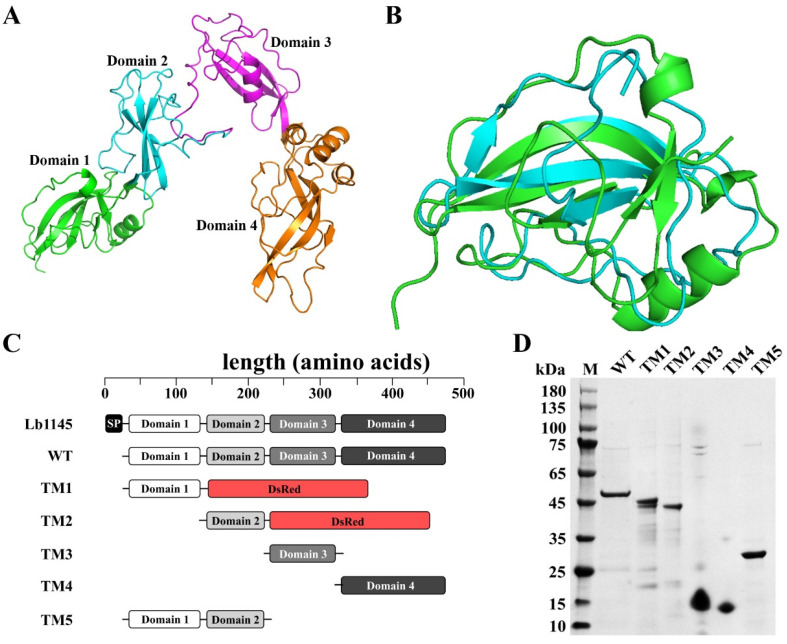
Delineation of Lb1145 into four domains. (**A**) The AlphaFold 3-predicted structure of Lb1145. (**B**) Structure superimposition of domain 1 and domain 2. Green: domain 1; Cyan: domain 2. (**C**) Schematic diagram showing the domain organization of Lb1145. (**D**) SDS-PAGE analysis of the wild-type and truncation mutant proteins of Lb1145. WT: wild-type protein; TM1 and TM2: the truncated derivatives of domain 1 and domain 2 in fusion with the red fluorescent protein DsRed, respectively; TM3 and TM4: the truncation derivatives of domain 3 and domain 4, respectively; TM5: the truncation derivative containing domain 1 and domain 2; DsRed: the red fluorescent protein.

**Figure 3 ijms-26-11612-f003:**
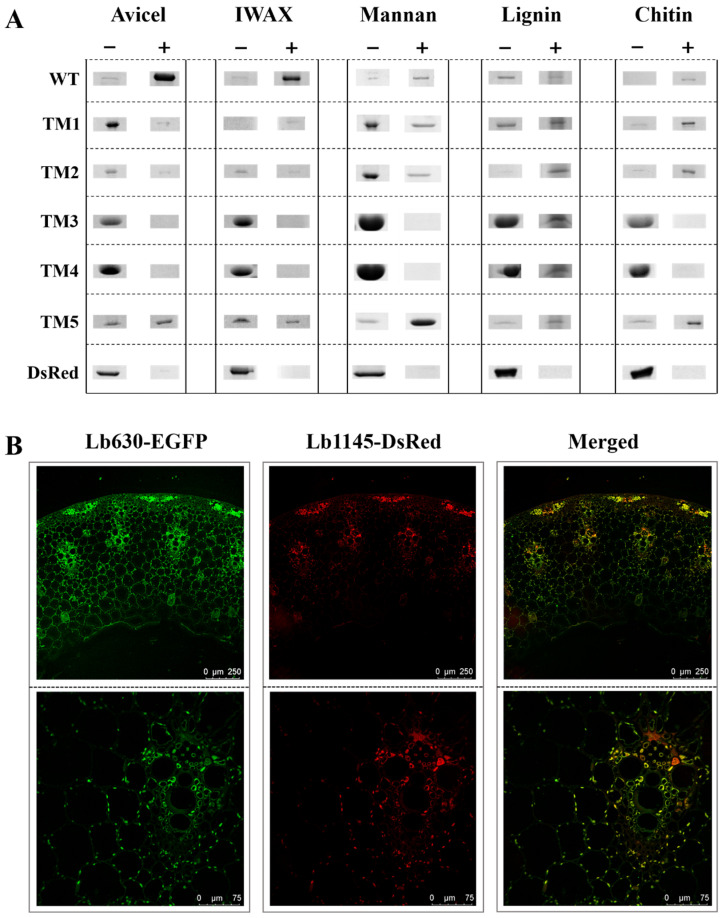
Binding of Lb1145 to plant cell wall. (**A**) Binding of the wild-type and its truncation mutants of Lb1145 to PCWPs, lignin, and chitin. Avicel: the crystalline cellulose. IWAX: insoluble wheat arabinoxylan; “−”: unbound protein; “+”: bound protein. (**B**) A comparison of the binding of Lb1145 and Lb630 to the wheat stem section. Lb1145-DsRed and Lb630-EGFP were incubated with a section of wheat stem, and the binding of the proteins was examined using a laser confocal microscope. Upper panels are the overall structure of the stem section. Lower panels are enlarged images of vascular bundle cells. Bars in upper and lower panels are 250 μm and 75 μm, respectively.

**Figure 4 ijms-26-11612-f004:**
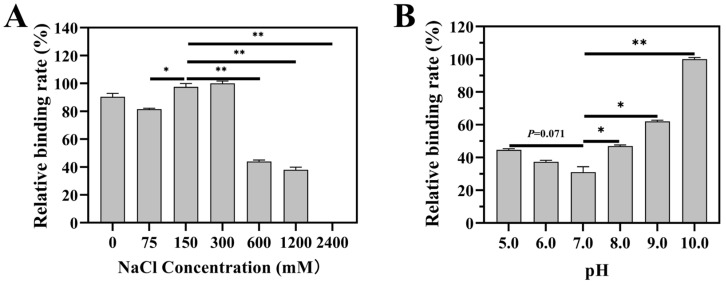
Effect of NaCl and pH on the binding of Lb1145 TM5 to cellulose. (**A**) NaCl concentration. (**B**) pH. Statistical significance was determined using one-way ANOVA. * *p*  <  0.05 and ** *p*  <  0.01; ns, no statistical significance; n  =  3.

**Figure 5 ijms-26-11612-f005:**
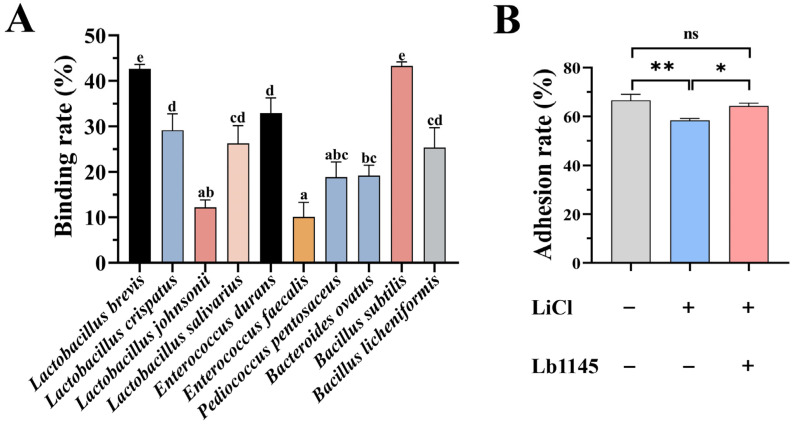
Lb1145 binds to selected intestinal bacteria and restores the adhesion of LiCl-treated *L. brevis* to intestinal epithelial cells IPEC-J2. (**A**) Binding of Lb1145 to selected intestinal bacteria. The Lb1145-DsRed protein (0.35 mg/mL) was mixed with equivalent bacterial density (OD_600_ = 1.5) of *L. brevis*, *L. mucosae*, *L. crispatus*, *L. johnsonii*, *L. reuteri*, *L. salivarius*, *E. durans*, *E. faecalis*, *P. pentosaceus*, *P. acidilactici*, *B. pseudolongum*, *B. longum*, *E. coli*, *B. uniformis*, *B. ovatus*, *B. subtilis*, and *B. licheniformis*, respectively, in a total volume of 250 μL. The mixture was incubated at 37 °C for 0.5 h. The fluorescence intensity of the supernatants after centrifugation was determined for calculating the relative adhesion rates. Only the bacteria with binding were displayed. Different letters (“a, b, c, d, and e”) on the columns indicate statistically significant differences among the bacteria (*p * <  0.05). (**B**) Lb1145 restored the adhesion of LiCl-treated *L. brevis* to the intestinal epithelial cells IPEC-J2. Statistical significance was determined using one-way ANOVA. * *p*  <  0.05 and ** *p*  <  0.01; ns, no statistical significance; n  =  3.

**Figure 6 ijms-26-11612-f006:**
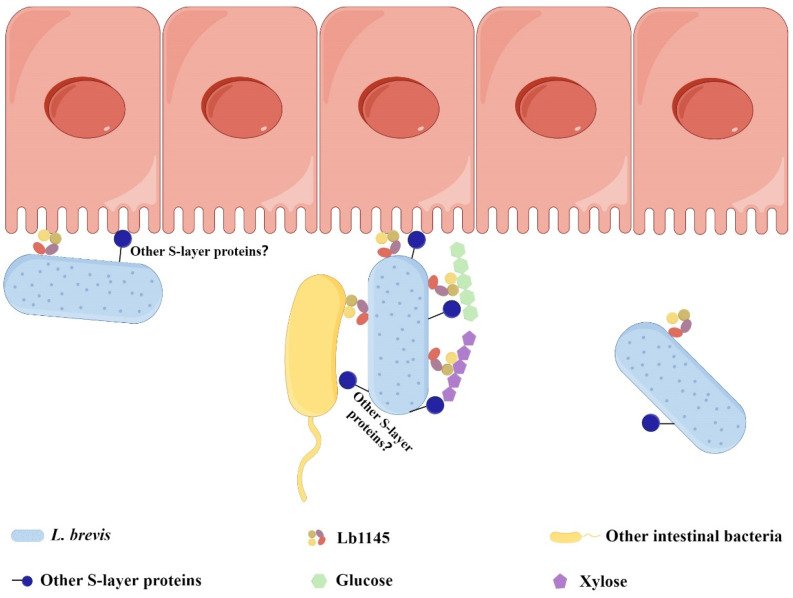
A simplified model depicting how the putative surface layer protein Lb1145 and the others may help *L. brevis* to adhere to PCWPs, other intestinal bacteria, and the intestinal epithelial cells. Note that only single polysaccharide chains of cellulose and xylan are displayed as representative PCWPs.

## Data Availability

The original contributions presented in this study are included in the article/Appendix A. Further inquiries can be directed to the corresponding author.

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
