# Peer review of "Nonspecific Binding of a Putative S-Layer Protein to Plant Cell Wall Polysaccharides—Implication for Growth Competence of *Lactobacillus brevis* in the Gut Microbiota"

_ijms, 2025, doi:10.3390/ijms262311612_

Round 1

Reviewer 1 Report

Comments and Suggestions for Authors

The manuscript titled “Nonspecific binding of an S-layer protein to plant cell wall polysaccharides – implication for growth competence of Lactobacillus brevis in the gut microbiota” by Hao, Z.; et al. is a scientific work where the authors addressed the interaction properties of plant cell wall binding proteins to plant polymers and xylan and cellulose. Some key parameters as the isoelectric point and the pH were assessed in this research. This is a topic of growing interest and the manuscript is generally well-written.

However, it exists some points that need to be addressed (please, see them below detailed point-by-point) to improve the scientific quality of the submitted manuscript paper before this article will be consider for its publication in the International Journal of the Molecular Sciences.

1) Introduction. “Plant cell wall polysaccharides (PCWPs) serve as an abundant but recalcitrant carbon source for many microbes living in the gut of the human and animals” (lines 39-40). Could the authors provide quantitative data insights according to the worldwide global burdens of intestinal diseases caused by gut microbiota alterations? This will significantly aid the potential readers to better understand the significance of this devoted research.

2) Results. “Using Avicel and insoluble WAX (…) discover that Lb636 (…) also bind to cellulose and xylan (…) The other proteins (…) could all bind to cellulose and xylan although the binding abilities were comparable weaker” (lines 123-128). Here, even if I agree with this statement provided by the authors, it should be also remarkable to discuss the high tendency of cellulose and xylan to uptake moisture decreasing their mechanical properties at the nanoscale [1] favouring the gut microbiota attack and digestion in the host organism [2].

[1] https://doi.org/10.1016/j.ijbiomac.2019.10.074

[2] https://doi.org/10.1016/j.fhfh.2025.100213

3) Did the authors test the effect of ionic strenght (in addition to the already examined protein isoelectric points) in the binding capabilities of polysaccharide binding proteins to plant cell wall polymers? This information is relevant to mimic their response inner the organism conditions.

4) Figure 4 (line 261). Some statistical analysis (e.g. One-way ANOVA as already detailed in the respective Materials & Methods section) needs to be conducted in order to discerned if the observed differences are statistically significant among the tested conditions. Same comment for the Fig. 5, panel A (line 287).

5) “3. Discussion” (lines 298-400). This section perfectly remarks the most relevant outcomes found by the authors in this work. The authors should consider to add a final “Conclusions” sections to shortly discuss about the promising future prospectives also highlighting the potential future action lines to pursue the topic covered in this research.

6) Materials & Methods. “4.2. Gene cloning, expression, and protein purification” (lines 419-457). Some information related to the protein purification should be added in an extra Figure to better visualize the yield in this process.

Reviewer 2 Report

Comments and Suggestions for Authors

Dear authors

The subject of the study is interesting and the manuscript was generally well writen. However, there are some issues that need to be addressed:

  • The authors should demonstrate that Lb1145 is expressed at the surface of L. brevis. Otherwise, most of the conclusions can be only considered speculative. 
  • The reference to the protocol for obtaining L. brevis strain transcriptomic data was not cited in the text.
  • The description of the protocol of "Binding of the recombinant L. brevis surface proteins to insoluble PCWPs, lignin, and chitin" should be improved. For instance, it is not clear how the samples corresponding to the negative(-) conditions were obtained. 
  • "Binding of Lb1145 to the wheat stem": please could you explain the correlation between these results and those shown in Figure 3A (if any)? Could you clarify the relevance of these results in the context of the intestine?
  • "Involvement of a non-enzymatic glycosylation-like process in binding of Lb1145 to PCWPs": considering the hypothesis and given the pI of TM5 around 9 or higher, would minimal binding not be expected at these pH values?
Comments on the Quality of English Language

The language needs some revision. 

Round 2

Reviewer 1 Report

Comments and Suggestions for Authors

The manuscript quality did not reach the high-standards of quality related to IJMS. For this reason, I recommend a transfer desk to a companion journal.

Reviewer 2 Report

Comments and Suggestions for Authors

Dear authors

Although the manuscript has been improved, some points have not been properly addressed, or maybe some comments have been misunderstood:

- The experiment conducted to demonstrate that Lb1145 is expressed on the surface of L. brevis actually only demonstrates that Lb1145 is capable of binding to the surface of L. brevis, as the authors themselves indicate in the caption of Fig. S1. So, most of the conclusions remain speculative.

- About the "Binding of Lb1145 to the wheat stem", it is not clear to me what is the correlation between those results and the adhesion ability of L. brevis in the context of the intestine. Please, clarify.

- About the effect of pH on the binding of TM5 to PCWPs:  since the hypothesis states that binding is mediated by hydrogen bonds and/or electrostatic forces, wouldn't we expect binding to be lower at pH values close to the pI of TM5?

Round 3

Reviewer 1 Report

Comments and Suggestions for Authors

The authors did a great deal of effort to cover all the suggestions raised by the Reviewers and the manuscript scientific quality was greatly improved. This work can be accepted for further publication in its current form.

Reviewer 2 Report

Comments and Suggestions for Authors

The main concerns have been addressed, The changes made contributed to improve the manuscript.